# Preparation and Characterization of a Glutathione-Responsive Doxorubicin Prodrug Modified by 2-Nitrobenzenesulfonamide Group—Its Selective Cytotoxicity Toward Cells with Enhanced Glutathione Production

**DOI:** 10.3390/ijms26094128

**Published:** 2025-04-26

**Authors:** Tomona Yukimura, Tomohiro Seki, Toshinobu Seki

**Affiliations:** Faculty of Pharmacy and Pharmaceutical Sciences, Josai University, 1-1 Keyakidai, Sakado, Saitama 350-0295, Japan; gyd2305@josai.ac.jp (T.Y.); sekt1042@josai.ac.jp (T.S.)

**Keywords:** CD44v, glutathione, drug release, prodrug, doxorubicin, DNA intercalation

## Abstract

GSH biosynthesis is enhanced in cancer cells that express the variant isoform of the surface antigen CD44 (CD44v), which is overexpressed in certain types of cancer. The GSH-responsive prodrug Ns-Dox was prepared by modifying the GSH-responsive group 2-nitrobenzene sulfonyl (Ns) with the model drug doxorubicin (Dox). Its function was evaluated based on its molecular interaction with model DNA in terms of its binding constant (*K_a_*). The association constant of Ns-Dox was lower, and its interaction with model DNA was weaker compared to that of Dox, suggesting that Ns-Dox may act as a less toxic prodrug. HCT116 cells with high CD44v expression and GSH levels and BT474 cells with low CD44v expression and GSH levels were used. The addition of Ns-Dox to HCT116 cells produced cytotoxic effects similar to those of Dox. In contrast, a significant difference in viability was observed between Ns-Dox- and Dox-treated BT474 cells at low concentrations. These findings suggest that Ns-Dox functions as a prodrug with low environmental toxicity and a lower GSH concentration in cancer cells. It is efficiently activated to Dox in cells with high GSH production, demonstrating its cell-killing effects.

## 1. Introduction

Over the past several decades, anticancer drugs have been extensively studied; however, conventional chemotherapeutic agents still suffer from low specificity toward cancer cells [1,2,3]. This lack of selectivity often leads to severe side effects in patients, posing a major challenge to the continued administration of treatment [3]. In response to these challenges, advanced drug delivery systems have been developed to enhance the efficacy and specificity of cancer treatment.

Stimuli-responsive drug activation systems, including prodrugs and targeting strategies that utilize materials recognizing specific receptors expressed on cancer cells, are expected to be a type of drug delivery system that reduces side effects in the pharmacotherapy of diseases, such as cancer [4,5,6]. Stimuli-responsive prodrugs for cancer therapy often utilize acidic pH, reducing conditions, and overexpressed enzymes [6,7,8]. Glutathione (GSH) has attracted attention as a stimulus for prodrug activation. It plays an important role in maintaining the redox balance of cells and acts as an antioxidant, protecting cells from oxidative stress [9,10]. Extracellular GSH concentrations in vivo, such as in blood, are estimated to be 2–20 μM, while intracellular GSH concentrations have been reported to range from 1–10 mM, indicating a markedly higher level within cells compared to the extracellular environment [9,10]. In some cases, such as colon and breast cancers, GSH concentrations have been observed to be 10 times higher than those in normal cells [11,12]. One of the factors that promotes GSH biosynthesis is the cystine/glutamate antiporter overexpression system x_c_- [13,14]. The central role of stem x_c_- is facilitated by the membrane transporter xCT, which provides intracellular stains for GSH biosynthesis. xCT is stabilized at the cell surface by a variant isoform of CD44 (CD44v), and CD44v-positive cancer cells show enhanced intracellular GSH biosynthesis (Appendix A) [15,16,17,18]. This environment is believed to be one reason why cells evade oxidative stress and enhance their resistance to radiotherapy and anticancer treatment.

In this study, we report the preparation of a prodrug that uses the high-GSH environment present in CD44v-positive cancer cells as a chemical stimulus to activate the drug [17,18,19,20,21,22,23]. As a GSH-responsive drug-activating prodrug, a GSH-responsive group, 2-nitrobenzenesulfonyl (Ns group), was modified on the amino group of the model anticancer drug doxorubicin (Dox) to be cleaved by the high levels of GSH present in cancer cells (Figure 1A). Many GSH-responsive prodrugs have been developed using compounds with disulfide bonds on the response-modifying side, but their extracellular stability has been questioned [19,20,21,22,23,24,25]. In contrast, the Ns group is stable in blood and extracellular environments with GSH concentrations of 2–20 µM, and some attempts using the Ns group have been reported for molecular probes and the intracellular delivery of RNA [23,24,25]. The Ns group is a functional group widely used in organic chemistry to protect primary amino groups, and its deprotection is achieved using thiolate nucleophiles [26]. The mechanism of deprotection of the Ns group is thought to involve the Meisenheimer complex, resulting from electron delocalization of the anionic intermediate by electron-withdrawing groups on the benzene ring after a nucleophilic addition reaction by the SH group of GSH (Figure 1B) [26,27]. Another advantage of the Ns group is that its reactivity with GSH can be adjusted by modifying the benzene ring with electron-withdrawing or electron-donating groups, as required. In this study, we designed Ns-Dox by introducing an electron-withdrawing carboxylic acid ester group into the benzene ring of the Ns group, which could be connected to CD44v-targeting nanocarriers in future studies.

To confirm that the prepared Ns-Dox releases Dox in response to GSH as a non-toxic prodrug, isothermal titration calorimetry (ITC) was employed to study its molecular interaction with DNA. Intermolecular interactions with DNA were examined. It has been suggested that the reduction in cytotoxicity due to the chemical modification of the primary amino groups of Dox by the prodrug is primarily attributed to changes in the molecular interactions with DNA [28,29,30]. As a proof of concept, we investigated whether Ns-Dox could be activated by chemical stimuli in an environment with controlled GSH concentrations [31]. To explore whether Ns-Dox could be activated as a prodrug by chemical stimuli in a controlled GSH environment, we investigated the GSH response activation of prodrugs in a non-cellular system. In vitro activation of Ns-Dox and its cytotoxic effects were examined in human colon cancer-derived HCT116 cells, which are reported to overexpress CD44v and xCT, and in human breast cancer-derived BT cells, which are reported to express both CD44v and xCT at lower levels [32,33,34,35,36]. GSH-responsive drug activation systems appear promising for selective cancer cell therapy and for reducing systemic side effects [18,25,37]. 

## 2. Results and Discussion

### 2.1. Synthesis of Ns-Dox

Ns-Dox was synthesized by modifying the Ns group with the primary amino group in the Dox amino sugar moiety (Figure 1). Fast atom bombardment mass spectrometry (FAB-MAS) and nuclear magnetic resonance (NMR) were performed to confirm the synthesis of the target substance. The molecular weight ion peak at 786.1572 for the target compound ([M]^+^ requires 786.1750) was confirmed using FAB-MS. Furthermore, NMR measurements of ^1^H, ^13^C, ^1^H-^1^H COSY, and ^13^C-^1^H HSQC confirmed the N-group modification of the amino group of Dox, as all peaks were assigned (Appendix A). Dox is a fluorescent anticancer drug, and its intrinsic fluorescence has been utilized in numerous past studies to investigate its release behavior from formulations and intracellular translocation. In this study, fluorescence spectroscopy was performed in order to examine changes in the fluorescence properties of doxorubicin caused by N-modification. The result of the fluorescence spectrum of Ns-Dox showed that the fluorescence originally exhibited by Dox (λ_ex_ = 480 nm, λ_em_ = 557 nm) and shown in Figure 2 was greatly suppressed. In the following study, we used this fluorescence property to evaluate the activation of Ns-Dox based on fluorescence intensity.

### 2.2. Molecular Interaction of Ns-Dox and Dox with DNA

The ability of Ns-Dox to function as a prodrug with lower cytotoxicity than that of Dox was investigated using DNA. The interaction of DNA with Dox and Ns-Dox was evaluated using ITC based on the binding constants (*K_a_*) and thermodynamic parameters. The results are summarized in Table 1. *K_a_* and binding enthalpy (Δ*H*) were calculated by fitting the experimental ITC values with a non-linear minimization method, as shown in Figure 3. The parameters of the free energy of bonding (Δ*G*) and the contribution of entropy change (−*T*Δ*S*) were calculated based on the following Equation (1) using standard thermodynamic relationships:Δ*G* = −*RTlnK* = Δ*H*−*T*Δ*S*(1)

The molar ratios of the resulting complexes were estimated from the ITC data, showing binding molar ratios of 9.5 for Dox to DNA and 8.7 for Ns-Dox to DNA. As shown in Table 1, the *K_a_* of Dox and Ns-Dox with DNA were 1.6 × 10^6^ M*^−^*^1^ and 1.0 × 10^5^ M*^−^*^1^, respectively, with Ns-Dox exhibiting approximately one-sixteenth the affinity of Dox. The Δ*H* values were −35.5 kJ/mol for Dox and −10.0 kJ/mol for Ns-Dox. It is well established that the primary mode of interaction between Dox and DNA involves intercalation between base pairs within the DNA minor groove, typically accompanied by a negative Δ*H* [38]. The relatively small negative Δ*H* and low *K_a_* observed for Ns-Dox are attributed to the loss of the positive charge on the basic nitrogen of Dox upon Ns modification, which reduces electrostatic interactions with the phosphate groups in DNA. Additionally, the introduction of the Ns group likely induces steric hindrance, further contributing to the weakened interaction [38,39]. These findings collectively indicate that Ns-Dox exhibits a lower affinity for DNA compared to Dox.

### 2.3. Evaluation of Ns-Dox Activation of the GSH Response in an In Vitro Non-Cellular System

To evaluate the GSH-responsive activation of Ns-Dox, Dox activation by Ns-Dox was investigated using a non-cellular in vitro system. Ns-Dox was incubated at a concentration of 20 µM in a buffer solution under conditions mimicking blood and extracellular GSH concentrations with 20 µM GSH, or under conditions simulating the high GSH concentration environment inside cancer cells with 1 and 5 mM GSH. The reaction solution was sampled over time, and the concentration of converted Dox was calculated based on its fluorescence intensity. As shown in Figure 4, Dox was activated at GSH concentrations of 1 and 5 mM in a GSH concentration-dependent manner. Cumulative activation rates of 11.4% and 40.7% were observed for 1 and 5 mM GSH, respectively, at 8 h. On the other hand, at 20 µM GSH, which mimics the extracellular GSH concentration, no activation to Dox was observed even 8 h after the addition of Ns-Dox, indicating that Ns-Dox is stable. These results suggest that Ns-Dox is not activated in environments with low extracellular GSH concentrations, such as the blood; instead, it is activated to Dox only in environments with high intracellular GSH concentrations.

### 2.4. Determination of CD44v and xCT Expression Levels and GSH Concentrations in Cells Utilized for the Evaluation of Ns-Dox Activation

To evaluate Ns-Dox activation in cells with different CD44v and xCT expression levels and GSH concentrations, we quantified CD44v and xCT expression and intracellular GSH levels in HCT116 cells, which have been reported to overexpress CD44v and xCT, as well as BT474/Cytomegalovirus-Luciferase (BT474) cells. The results of immunofluorescence analysis using confocal laser scanning microscopy confirmed the co-expression of CD44v and xCT in HCT116 cells (Appendix A). Quantification of the immunofluorescence using flow cytometry, as shown in (Appendix A), yielded consistent results, confirming that the expression levels of CD44v and xCT in HCT116 cells were higher than those in BT474 cells. Intracellular GSH levels were quantified using the GSH/GSSG assay, which allows the fractional quantification of GSSG, the oxidized form of GSH. The GSH concentration was 64.6 nmol/mg protein in HCT116 cells and 22.1 nmol/mg protein in BT474 cells, representing a 3-fold difference in concentration (Appendix A). These results suggest that in HCT116 cells overexpressing both CD44v and xCT, GSH biosynthesis is promoted and the intracellular GSH concentration is higher, as shown previously [16,18,40].

### 2.5. Quantitative Evaluation of the Cellular Uptake of Ns-Dox and the Activation of GSH Response In Vitro

Two types of live cells, differing in CD44v and xCT expression levels as well as GSH concentrations, were employed to investigate the uptake of Ns-Dox into cells and its activation to Dox. HCT116 and BT474 cells were added to phenol red-free medium containing 20 µM of either Ns-Dox or Dox and then incubated for various durations. The amounts of Ns-Dox and Dox in the cells were quantitatively evaluated by high-performance liquid chromatography (HPLC). Bars A and B in Figure 5 show the amount of drug detected per number of cells obtained within a cell over time. In the presence of Ns-Dox, the amount of Ns-Dox detected inside the cells is shown in blue, and the amount of Dox taken up by the cells as Ns-Dox and activated (activated from Ns-Dox) is shown in orange. The Dox addition conditions are shown by red bars, indicating the amount of Dox detected in the cells. The cellular uptake of Dox increased over time in both cell types and was consistently higher than the total dose of Ns-Dox and Dox under Ns-Dox conditions at all time points. The reduced intracellular uptake of Ns-Dox compared to Dox can be attributed to the following two factors: decreased membrane permeability due to the increased molecular weight of Ns-Dox (786.72) relative to Dox (543.52) and reduced transport via organic anion-transporting polypeptides (OATPs), possibly due to the loss of the positive charge on the amino sugar moiety of Dox following Ns modification [41,42]. Consequently, these factors are hypothesized to collectively contribute to the lower cellular uptake of Ns-Dox compared to Dox. The percentage of Ns-Dox taken up as Ns-Dox and activated to Dox is shown as the activation rate in the graph in Figure 6. The activation ratio of Ns-Dox in HCT116 cells with a high GSH concentration was 23.2% at 6 h, whereas the activation ratio of Ns-Dox in BT474 cells with a low GSH concentration was 9.2% at 6 h, indicating that the activation ratio of Ns-Dox in HCT116 cells was higher than that in BT474 cells. Based on these results, it was expected that Ns-Dox would be activated more efficiently in cells with higher concentrations of GSH.

### 2.6. Evaluation of the Subcellular Distribution of Activated Dox After the Addition of Ns-Dox Using Confocal Laser Scanning Microscopy

Quantitative evaluation of GSH response activation following Ns-Dox intracellular uptake in an in vitro cell system confirmed that Ns-Dox was taken up intracellularly and activated into Dox. Confocal laser scanning microscopy was used in order to gain further insight into the intracellular distribution of activated Dox. In HCT116 cells, an increase in Dox fluorescence intensity was observed over time after the addition of Ns-Dox compared to the control, which contained only phenol red-free medium, as shown in Figure 7. The activated Dox was predominantly distributed in the cytoplasm between 2 and 6 h. At 24 h after the addition of Ns-Dox, Dox was distributed throughout the nucleus. The distribution was similar to that observed at 2 h following the addition of 20 µM Dox, as shown in Appendix A. After changing from Ns-Dox to Dox, the accumulation of Dox in the nucleus, the site of action, was evident. By contrast, Dox fluorescence was not observed in BT474 cells within this time range, indicating that Dox was not activated in cells with low GSH levels. These results were consistent with the results of the quantitative cellular uptake assessment of the GSH-responsive activation of Ns-Dox in vitro.

### 2.7. Evaluation of Ns-Dox Cytotoxicity

The cytotoxicity of Ns-Dox was evaluated based on cell viability 24 h after the addition of Ns-Dox or Dox in HCT116 cells with high CD44v and xCT expression, as well as GSH biosynthesis, and in BT474 cells with low levels of these markers. As shown in Figure 8A, HCT116 cells exhibited similar cell-killing effects with Ns-Dox and Dox at each tested concentration. In contrast, as depicted in Figure 8B, the survival ratios of BT474 cells under 100 µM conditions for Ns-Dox and Dox were 77.9% and 9.0%, respectively. In the 0.05 µM condition, the survival ratios were 101.3% and 84.3%, respectively, indicating a notable difference in the low-concentration conditions. Cell viability also modestly decreased with increasing concentrations of Ns-Dox in BT474 cells but remained higher at all concentrations compared to HCT116 cells.

The IC_50_ values were analyzed from these results using Kaleida Graph. The IC_50_ of Ns-Dox in BT474 cells could not be calculated due to its high viability beyond the tested range. The IC_50_ values of Ns-Dox and Dox in HCT116 cells were 8.5 µM and 8.2 µM, respectively, indicating nearly identical drug effects. On the other hand, the IC_50_ of Dox in BT474 cells was 5.4 µM, indicating that BT474 cells are more sensitive to Dox than HCT116 cells. Nevertheless, the consistently high viability of BT474 cells across all tested concentrations of Ns-Dox suggests that Ns-Dox exhibits reduced cytotoxic activity, even in cells that are otherwise sensitive to Dox, potentially reflecting lower activity in normal cells as well. Based on the above, Ns-Dox may function as a prodrug with low toxicity in an environment with low GSH concentration in cancer cells and be efficiently activated to Dox in cells with high GSH production, resulting in a cell-killing effect. However, the observed cytotoxicity of Ns-Dox in each cell line should be interpreted while considering multiple factors, including the cellular uptake of Ns-Dox, its intracellular conversion to Dox, and the inherent sensitivity of the cells to Dox. As shown in Figure 5, the quantitative analysis of cellular uptake revealed comparable levels of Dox and Ns-Dox in both BT474 and HCT116 cells. Furthermore, as illustrated in Figure 8, HCT116 cells exhibited lower sensitivity to Dox compared to BT474 cells. As shown in Appendix A, the intracellular GSH level in HCT116 cells was approximately three times higher than in BT474 cells, which likely contributed, at least in part, to the more efficient conversion of Ns-Dox to Dox and the resulting differences in cytotoxicity. However, the fact that Ns-Dox exhibited a two-orders-of-magnitude difference in IC_50_ values between the two cell lines suggests that additional factors not explored in the present study may also be involved.

## 3. Materials and Methods

### 3.1. Materials

Potassium dihydrogen phosphate, disodium hydrogen phosphate, anhydrous magnesium sulfate (MgSO_4_), sodium chloride (NaCl), cesium carbonate, magnesium oxide (MgO), hydrochloric acid (HCl), tetrahydrofuran (THF), *N,N*-dimethylformamide (DMF), methanol (MeOH), *n*-Hexane, dichloromethane (DCM), acetonitrile (MeCN), ethyl acetate (EtOAc), dimethyl sulfoxide (DMSO), chloroform, reduced glutathione, and 5-sulfosalicylic acid dihydrate (SSA) were supplied by Fujifilm Wako Pure Chemical Industries (Osaka, Japan). Methyl 4-(chlorosulfonyl)-3-nitrobenzoate, glycylglycine, and *N*-chlorosuccinimide were purchased from the Tokyo Chemical Industry Co. Potassium chloride (KCl) was purchased from Kanto Chemical Co. Thioacetic acid and Triton X-100 were purchased from Sigma-Aldrich (St. Louis, MO, USA). Doxorubicin hydrochloride was purchased from Carbosynth (San Diego, CA, USA), along with radioimmunoprecipitation assay (PIRA) buffer (Thermo Fisher Scientific, Waltham, MA, USA).

### 3.2. Cell Line

Human colon cancer-derived HCT116 cells (RCB2979) were obtained from the RIKEN BRC Cell Bank (Ibaraki, Japan), while human breast cancer-derived BT474/CMV-Luc cells (JCRB1450) were obtained from the Human Science Resource Bank (Osaka, Japan). The cells were grown in Dulbecco’s modified Eagle’s (DMEM) medium, Roswell Park Memorial Institute (RPMI-16) medium, 10% fetal bovine serum (FBS), 1% penicillin-streptomycin solution, 4% paraformaldehyde phosphate buffer (PFA), bovine serum albumin (BSA), and 0.25% trypsin/0.53 1 mM EDTA, which were obtained from Fujifilm Wako Pure Chemical Industries, Ltd. HCT116 cells were cultured in DMEM medium, while BT474 cells were cultured in RPMI-16 medium with FBS and 1% penicillin-streptomycin solution at 37 °C in a humidified atmosphere of 5% CO_2_.

### 3.3. Methods

#### 3.3.1. Synthesis of the Prodrug

Synthesis of methyl 4-(chlorosulfonyl)-3-nitrobenzoate (Ns-Cl) Methyl 4-(chlorosulfonyl)-3-nitrobenzoate was carried out as reported [25]. Glycylglycine (130 mg, 1 mmol) was dissolved in dehydrated DMF (20 mL). Thioacetic acid (0.81 mL, 11 mmol) and cesium carbonate (7.30 g, 22 mmol) were added to the glycylglycine solution under ice-cold conditions, followed by stirring for 10 min. Methyl 4-chloro-3-nitrobenzoate (1.61 g, 7.5 mmol) was added to the reaction solution, and then the reaction solution was stirred overnight on ice in the dark. EtOAc and a 1 M hydrochloric acid aqueous solution were added, and the organic phase was washed with distilled water and brine. The organic layer was dried over MgSO_4_, and the solvent was removed using a rotary evaporator to obtain a white powder. The resulting powder was then dissolved in 10 mL MeCN and NCS (3.90 g, 29 mmol) was dissolved in 24 mL MeCN containing 2 M HCl. The reaction solution was stirred at 0 °C for 1 h. The solvent was then removed using a rotary evaporator to obtain the crude product. Thin-layer chromatography (TLC) confirmed the Rf values of 0.31 for the target product (Ns-Cl), 0.13 for the SH intermediate, and 0.58 for the starting material in *n*-Hexane/DCM (1/1). The crude product was purified by gradient flash column chromatography using an *n*-Hexane/DCM mobile phase, followed by the evaporation of the eluent to obtain 4-(chlorosulfonyl)-3-nitrobenzoate methyl as a yellowish powder (1.02 g, 49% yield).

The synthesis of 2-nitrobenzenesulfonyl doxorubicin (Ns-Dox) was conducted as follows: The synthesis of 2-nitrobenzenesulfonyldoxorubicin (Ns-Dox) was conducted as previously described [43]. After dissolving Dox (290 mg, 0.5 mmol) in a THF/H_2_O (3/1) mixture (22.5 mL THF/7.5 mL H_2_O), MgO (180.6 mg, 2.5 mmol) was added. The reaction was initiated by dissolving Ns-Cl (230.7 mg, 0.83 mmol) in 7.5 mL of dehydrated THF and dropping it into the Dox solution while stirring vigorously at 25 °C, and the reaction was followed by TLC. Chloroform/MeOH (95/5) gave an Rf value of 0.25 for the target (Ns-Dox), with the starting material being the origin. After 2.5 h, MgO was removed using filter paper, and 150 mL of EtOAc was added, washed with 50 mL of water, dehydrated with MgSO_4_, and concentrated. The product was charged onto a silica gel column and prepared using a mobile phase of chloroform/MeOH based on the TLC information. The solvent in the target fraction was removed to obtain a purified red product (235.2 mg, 60% yield).

NMR spectroscopy was performed using an AVANCE NEO 400 system (Bruker Japan, Inc., Billerica, MA, USA) and an MS JMS-700 (JEOL Corporation, Tokyo, Japan). ^1^H NMR (400 MHz, DMSO-*d*_6_):δ 13.92 (1H, s), 13.20 (1H, s), 8.26–8.25 (1H, s, *J* = 1.6 Hz), 8.15–8.14 (1H, dd, *J* = 6.6, 1.6 Hz), 8.15–8.09 (2H, m) 7.89–7.86 (1H, *J* = 7.6 Hz) 7.86–7.83 (1H, dd, *J* = 5.8, 2.0 Hz), 7.63–7.60 (1H, dd, *J* = 6.0, 2.0 Hz), 5.41 (1H, s), 5.15 (1H, d), 4.98 (1H, d, *J* = 6.0 Hz), 4.90–4.87 (1H, t, *J* = 4.7 Hz), 4.85–4.82 (1H, t, *J* = 6.0 Hz), 4.54–4.53 (2H, d, *J*= 6.0 Hz), 4.10–4.05 (1H, q, *J* = 6.6 Hz), 3.96 (3H, s), 3.74 (3H, s), 3.59 (1H, br), 3.59 (1H, overlaps with H_2_O), 2.10 (2H, m), 1.95–1.87 (1H, t-d, *J* = 9.1, 3.9, 3.8, 3.8 Hz), 1.43–1.38 (1H, dd, *J* = 8.0, 4.6 Hz), 1.08–1.06 (3H, d, *J* = 6.5 Hz). ^13^C NMR (100 MHz, DMSO-*d*_6_): δ 214.10, 186.93, 186.85, 163.86, 161.20, 156.45, 154.80, 147.67, 138.23, 135.85, 135.07, 136.67, 134.53, 134.42, 133.21, 130.64, 125.16, 120.40, 120.11, 119.41, 111.15, 111.80, 100.17, 75.38, 70.30, 69.16, 67.11, 64.06, 57.01, 50.77, 37.20, 32.58, 30.79, 17.37. HRMS (FAB, positive mode, matrix: 3-nitrobenzyl alcohol) *m*/*z* 786.1572 ([M^+^] required 786.1750) C_35_H_34_N_2_O_17_S.

The fluorescence spectra of the obtained Ns-Dox were examined using an RF-5300PC fluorescence spectrophotometer (Shimadzu Corporation, Kyoto, Japan). Here, 200 nM Dox or Ns-Dox were measured at λ_ex_ = 480 nm and λ_em_ = 400–800 nm, with bandwidths of λ_ex_: 5 nm and λ_em_: 5 nm and a sampling interval of 1 nm. Transparent quartz cells (Sansho Corporation, Tokyo, Japan) were used to measure 200 nM Ns-Dox and Dox in 100 mM phosphate buffer (PBS, pH 7.4) at room temperature.

#### 3.3.2. Assembly of Model DNA

Two oligonucleotides of twenty base pairs (5′-CGT ACG CGT ACG CGT ACG CG-3′ and 5′-CGC GTA CGC GTA CGC GTA CG-3′) were purchased from the FASMAC Corporation (Tokyo, Japan). Tris-HCl buffer (pH 8.0) was purchased from Nippon Gene (Tokyo, Japan). Two oligonucleotides of twenty base pairs, namely 5′-CGT ACG CGT ACG CGT ACG CG-3′ and 5′-CGC GTA CGC GTA CGC GTA CG-3′, were incubated at a 200 µM concentration mixed in 10 mM Tris-HCl buffer (pH 7.4). Annealing and hybridization were performed by cooling the mixture to 95 °C for 5 min, 50 °C for 30 min, and 4 °C for 20 min using a 100TM Thermal Cycler Gradient device (Bio-Rad Laboratories, Hercules. CA, USA) [44]. Double-stranded DNA formation was confirmed using a J-1500-150DS circular dichroism spectrometer (Japan Spectroscopic Corporation, Tokyo, Japan). The measurement range was 200–600 nm, the bandwidth was 1 nm, and data acquisition was carried out at 0.5 nm at 25 °C to confirm double-stranded DNA formation.

#### 3.3.3. Interaction with DNA

ITC experiments were performed at 25 °C using a Nao ITC isothermal titration calorimeter (TA Instruments, New Castle, DE, USA). All solutions were degassed in a degassing station (TA Instruments) to prevent bubble formation. Dox, Ns-Dox, and DNA were dissolved directly in 10 mM Tris-HCl buffer (pH 7.4). The experiment was conducted as follows: first, 2.5 µL (20 injections) of Dox (200 µM) or Ns-Dox (200 µM) solution was added to the side of the syringe. A solution of DNA in Tris-HCl buffer (0.004 mM) was added to the cell side [31]. A blank was generated by injecting the titrant into a cell filled with the Tris-HCl buffer. This blank was subtracted from the corresponding titration to account for the dilution heat. The titrants were injected approximately 200 sec apart, with each injection lasting 20 sec. The stirring speed was maintained at 350 rpm to ensure uniform mixing in the cell. The peaks were integrated and corrected for dilution effects to obtain a combined isotherm. All data were analyzed using NanoAnalyze v. 3.1.2 (TA Instruments) provided by the manufacturer. An “independent model” was used to evaluate the results.

#### 3.3.4. Evaluation of the GSH Response Activation of Ns-Dox in an In Vitro Non-Cellular System

The conversion of Ns-Dox to Dox was measured using a fluorescence spectrophotometer with λ_ex_ = 480 nm, λ_em_ = 400–800 nm, bandwidths λ_ex_: 5 nm, λ_em_: 5 nm, and a sampling interval of 1 nm. Activation of Ns-Dox was assessed by incubating it at a final concentration of 20 µM in PBS containing GSH at concentrations of 20 µM and 1 mM, and 5 mM. Samples were incubated at 37 °C and collected at various time points over 0–8 h. The extent of Dox activation was evaluated by monitoring the fluorescence intensity derived from Dox (λ_em_ = 557 nm).

#### 3.3.5. Immunofluorescence Analysis by Laser Scanning Confocal Microscopy

Cells (2 × 10^4^ cells/well) were seeded onto poly-l-lysine-coated culture coverslips (Matsunami Glass Co., Ltd., Osaka, Japan) in 12-well plates and incubated overnight at 37 °C in 5% CO_2_. After washing, the cells were fixed in 4% PFA for 15 min, incubated in staining buffer (3% BSA-PBS) containing 0.2% Triton X-100 for 30 min at room temperature, and washed with PBS. After incubation in staining buffer (3% BSA-PBS) for 60 min at room temperature, the cells were incubated in staining buffer (1% BSA-PBS) diluted with primary antibodies (anti-human CD44v9-rat IgG2_a_ (1:33) (Cosmo Bio Inc., Tokyo, Japan) and anti-xCT-rabbit IgG polyclonal (1:20) (Abcam, Cambridge, UK)) at 4 °C overnight. The cells were washed three times with PBS and then incubated in staining buffer (0.2% BSA PBS) with the appropriate goat anti-rabbit IgG Texas Red-labeled (1:500) and goat anti-rabbit IgG Alexa Fluor 488-conjugated (1:500) secondary antibodies (Invitrogen, Waltham, MA, USA) for 3 h at 4 °C. After washing three times with PBS, the cells were placed on Vectashield mounting medium drops containing 4′,6-diamidino-2-phenylindole (DAPI), with the cell adhesion surface facing down. The FV1000-D confocal laser scanning microscope (Olympus, Tokyo, Japan) was used to measure Texas Red as follows: λ_ex_ = 561 nm, λ_em_ = 570–670 nm (CD44v—red), Alexa Fluor 488 λ_ex_ = 488 nm, λ_em_ = 500–540 nm (xCT—green), and DAPI λ_ex_ = 405 nm, λ_em_ = 430–470 nm (Nuclear—blue). To check the specificity of the antibody, a similar procedure was performed with a rat IgG isotype control (Invitrogen, Waltham, MA, USA) and a rabbit IgG isotype control (Novus Biologicals, Centennial, CO, USA) for the primary antibodies (Appendix A).

#### 3.3.6. Immunofluorescence Analysis by the Flow Cytometer

Cells (1 × 10^6^ cells/well) were seeded in 6-well plates and incubated overnight at 5% CO_2_, 37 °C. Cells were digested with 0.25% trypsin/0.53 1 mM EDTA for 5 min and detached from the plates. The cell concentration was adjusted to 3 × 10^5^ cells/tube using staining buffer (0.2% BSA-PBS), and the primary antibodies anti-human CD44v9-Rat IgG2_a_ (1:33) and anti-human xCT-Rat IgG (1:100) were diluted in staining buffer (0.2% BSA-PBS), added to separate tubes, and incubated at 4 °C for 45 min. Cells were then centrifuged at 1500 rpm, washed twice with PBS, diluted with the secondary antibody goat anti-Rat IgG Texas Red (1:500) in staining buffer (0.1% BSA PBS), and incubated in the dark at 4 °C for 30 min. PBS was added to each tube, and the tubes were washed by centrifugation at 1500 rpm. Finally, the cells were resuspended in PBS and analyzed by flow cytometry (BECKMAN COULTER, Tokyo, Japan) using the CytExpert software ver. 2.0 with laser-Blue, FITC 525/40 nm. The data are based on the average fluorescence signal from 100,000 cells. To check the specificity of the antibody, a similar operation was performed using the antibody Rat IgG isotype control (Invitrogen) as the primary antibody (Appendix A).

#### 3.3.7. GSH/GSSG Assay

The assay was performed using a GSSG/GSH quantification kit (DOJINDO Laboratories, Japan) according to the manufacturer’s instructions. Cells were lysed with 0.25% trypsin/0. 53 1 mM EDTA for 5 min, detached from the plate, lysed by adding 100 µL of PIRA buffer per 1 × 10^7^ cells, mixed with 5% SSA lysis buffer, then centrifuged at 1500 rpm for 15 min at 4 °C before the supernatant was collected. The supernatant was collected and analyzed by measuring the absorbance at 405 nm with a BioTek Cytation microplate reader (Agilent Technology Corporation, Tokyo, Japan), and the value was corrected for protein quantification using the bicinchoninic acid (BCA) method.

#### 3.3.8. Quantitative Evaluation of Cellular Uptake of Ns-Dox and Activation of the GSH Response In Vitro

Cells were incubated overnight at 5% CO_2_, 37 °C. After washing with PBS, Dox or Ns-Dox 20 µM was added and incubated every hour at 5% CO_2_, 37 °C. After washing with PBS, the cells were digested with 0.25% trypsin/0.53 1 mM EDTA for 5 min. Next, 1 × 10^7^ cells/100 µL PIRA buffer was added to disrupt the cells, centrifuged at 1500 rpm for 15 min at 4 °C, and the supernatant was collected. The sample was diluted 2-fold and quantified using HPLC (fluorescence detector: FP-4025, column oven: CO-4061, autosampler: AS-4150, pump: PU-4180, Japan Spectroscopic Corporation, Tokyo, Japan).

The excitation wavelength was 480 nm, the fluorescence wavelength was 557 nm, and mobile phase A consisted of acetonitrile/water (30:70) and phosphoric acid (pH 3.0). Mobile phase B consisted of acetonitrile (100). The following gradient was used for mobile phase B: 15% 0 min, 100% 20 min. The column was an octadecylsilane (C18) column (Mightysil RP-18 GP 150-2.0 3 µm, Kanto Chemical Co., Ltd.) and the guard column was a Mightysil RP-18 GP 5-2.0 3 µm (Kanto Chemical Co., Ltd.). The temperature was 25 °C and the flow rate was 0.2 mL/min. The measurement data were analyzed using Chrom NAV Ver. 2 (Japan Spectroscopic Corporation, Tokyo, Japan).

#### 3.3.9. Evaluation of Subcellular Distribution of Ns-Dox by Confocal Microscopy

Cells (4 × 10^4^ cells/well) were seeded on poly-l-lysine-coated culture coverslips (Matsunami Glass Co., Ltd., Osaka, Japan) in 12-well plates and incubated overnight at 5% CO_2_, 37 °C. After washing with PBS, 20 µM Dox or Ns-Dox was added at each time point, and the cells were fixed in 4% PFA for 15 min and placed on a drop of Vectashield mounting medium containing DAPI with the cell adhesion surface facing down. Cells were observed under an FV1000-D confocal microscope (Olympus Corporation, Tokyo, Japan) with Alexa Fluor 594 (Dox—red) λ_ex_ = 561 nm, λ_em_ = 570–670 nm; (Nuclear—blue) λ_ex_ = 405 nm, λ_em_ = 430–470 nm; scale 100 µm.

#### 3.3.10. Evaluation of the Subcellular Distribution of Activated Dox After the Addition of Ns-Dox Using Confocal Laser Scanning Microscopy

To determine the cytotoxicity of the prodrugs in the cell lines, the MTT assay kit (Nacalitescu Corporation, Kyoto, Japan) was used according to the manufacturer’s instructions. Briefly, 96-well plates were seeded at 1 × 10^4^ cells/well for HCT116 cells and 1.5 × 10^4^ cells/well for BT474 cells. After washing with PBS, the cells were exposed to various concentrations (0.001~100 µM) of Dox or Ns-Dox in phenol red-free medium for 12 h. After 24 h, the medium was removed, and the cells were incubated for 1 d. The cells were incubated with MTT solution for 4 h at 37 °C. The formazan crystals were dissolved in 2-propanol. The absorbance was measured using a spectrophotometer, and cell viability was determined by subtracting the reference absorbance at 650 nm from the absorbance at 590 nm. IC_50_ was analyzed by fitting the data with a dose–response logistic regression curve model using KaleidaGraph version 4.01.

## 4. Conclusions

In this study, the amino group of Dox was chemically modified with the Ns group to prepare a prodrug, Ns-Dox, which is active in response to high concentrations of GSH. Ns-Dox was expected to function as a prodrug with a lower affinity for DNA and lower toxicity when compared to Dox based on the ITC measurements. The activation of Ns-Dox in in vitro non-cellular systems was found to be GSH concentration-dependent at 1 and 5 mM GSH. In contrast, at 20 µM of GSH, which mimics extracellular GSH concentrations, such as those found in blood, Ns-Dox was not activated, indicating that it can exist stably in extracellular environments, such as in blood.

The quantitative evaluation of Ns-Dox intracellular uptake and GSH response activation in in vitro cell systems revealed that HCT116 cells, which have high GSH concentrations, exhibited a greater ratio of Ns-Dox activation compared to BT474 cells, which have low GSH concentrations. In contrast, BT474 cells showed higher cell viability under Ns-Dox conditions than under Dox conditions. These findings suggest that Ns-Dox behaves as a prodrug with a low toxicity in environments with low intracellular GSH concentrations and is efficiently activated to Dox in cells with high GSH production, where it exerts cell-killing effects. The results of this study demonstrated that Ns-Dox was stable in extracellular environments, such as blood, and that it was successfully activated in cancer cells, where high GSH concentrations triggered its conversion. Future studies combining the Ns-Dox system with CD44v-targeted carriers may enhance cancer-selective therapy while reducing side effects.

## Data Availability

The original contributions presented in this study are included in the article/Appendix A. Further inquiries can be directed to the corresponding author.

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
