# Peer review of "Preparation and Characterization of a Glutathione-Responsive Doxorubicin Prodrug Modified by 2-Nitrobenzenesulfonamide Group—Its Selective Cytotoxicity Toward Cells with Enhanced Glutathione Production"

_ijms, 2025, doi:10.3390/ijms26094128_

Round 1

Reviewer 1 Report

Comments and Suggestions for Authors

The study is performed at high experimental level, its findings are of potential interest, however, a number of shortcomings is present.

     The difference in cL50 between the compounds (close to two orders of magnitude) is attributed to just 3x lower concentration of GSH in BT474 cells (Fig. 8B) compared to HCT cells, where there is no difference in cL50. In quantitative terms this is a doubtful explanation. Other mechanisms may have been evaluated or mentioned by the authors. In this context, the quality of Fig. 7 could be improved. A correlation between differences in GSH concentration and cL50 differences can only be made from a larger number of data with different cell types, different uptake etc. 

     A minor statement - #208-209 - why membrane permeability and not a transporter? Dox loses its positive charge during the modification.

    There are also numerous problems with the stylistics and comprehensibility of the mnuscript. For example, Abstract: #14 - kinds of therapy may be specified, or the statement discarded; #18: ... and dsDNA - misleading; #19-20: incomprehensible statement about low toxicity; Introduction is too long and vague, especially its last paragraph, where the over-describing misses the point.  It contains excessive general statements, which are insufficiently related to the topic of the study (#32-35, 39-43), unclear - 1-10 mM higher (#48), redundant (#51-52,64-66), erroneous - ... modified on... (#62), etc. In Results and Discussion,  #139-140,147-148 repeat the data of Table 1. #145-146- unclear, #166-172 - this part may be significantly shortened; #198 - phenol red; 

Comments on the Quality of English Language

It is advised to consult good/native English speaker.

Reviewer 2 Report

Comments and Suggestions for Authors

Dear Authors,

The manuscript - 3464189 titled “Preparation and Characterization of a Glutathione-Responsive Doxorubicin Prodrug Modified by 2-Nitrobenzenesulfonamide Group- its Selective Cytotoxicity toward Cells with Enhanced Glutathione Production” delineates study findings concerning the synthesis of doxorubicin prodrug and evaluation of its cytotoxicity against HCT116 and BT747 cells.

The title is informative and concise.

The abstract summarizes the main points of the paper.

The manuscript is well-structured and comprises the following sections: Abstract, Introduction, Results and Discussion, Materials and Methods, and Conclusions.

The introduction provides general information for stimuli responsive drug activation systems and the role of glutathione as stimulus for prodrug activation. The authors explain very well the choice of prodrug and cell lines for research purposes.

The methods used are appropriate for the study and detailed description is provided.

The results are described in detail and clearly presented in the main text and in the supplementary files. They are thoroughly analyzed discussed in the main text.

The conclusions are rational and based on the article's content. The referenced literature is appropriate and relevant to this research.

I have following remarks.
1) there are duplicate sentences in lines 51, 52

2) Reference 42 – needs correction, missing journal, pages, doi

In the iThenticate report 24% duplicate was reported. I reviewed the report very carefully and find that the repetitions are due to the use of well-known terms to describe the methodologies, names of reagents and equipment necessary for the analyses, which are available and used by various research groups.

Overall, the study is well designed and conducted and the findings demonstrated that doxorubicin prodrug studied is safe and stable in extracellular environment and was successfully activated in cancer cells with high GSH concentration. The article could generate ideas for developing drug delivery systems with fewer side effects. This gives me reason to recommend the article for publication in International Journal of Molecular Science after minor revision.

Round 2

Reviewer 1 Report

Comments and Suggestions for Authors

Please improve the quality of Figs. 7, S5A and S6A. 

Author Response

Thank you very much for your comments. We highly appreciate your review of the manuscript. According to your valuable suggestion, we revised the manuscript carefully. We would be grateful if you could kindly review our manuscript once again.

For the confocal laser scanning microscopy (CLMS)images, all imaging parameters—including exposure time and gain—were kept consistent across the datasets being compared, and the images were acquired at a resolution of 1024 pixels. Therefore, we consider the original image quality to be sufficient.

However, during the insertion of the images into the main text, the file format was unintentionally converted from TIFF to PNG, resulting in a noticeable reduction in image quality.

In this revision, we have ensured that all images in both the main manuscript and the supplementary materials remain in the original TIFF format.

While the image quality appears to have improved when viewed in the Word file we submitted, the issue remains unresolved in the manuscript PDF that we converted.

We apologize for the inconvenience, but when reviewing the CLMS images, please refer to the Word file. If the quality is still insufficient, individual image files have also been attached for your reference.